# One-Week Elderberry Juice Treatment Increases Carbohydrate Oxidation after a Meal Tolerance Test and Is Well Tolerated in Adults: A Randomized Controlled Pilot Study

**DOI:** 10.3390/nu15092072

**Published:** 2023-04-25

**Authors:** Bret M. Rust, Joseph O. Riordan, Franck G. Carbonero, Patrick M. Solverson

**Affiliations:** 1Department of Nutrition and Exercise Physiology, Elson S Floyd College of Medicine, Washington State University, Spokane, WA 99202, USA; brrust@iu.edu (B.M.R.); joe.riordan@wsu.edu (J.O.R.); franck.carbonero@wsu.edu (F.G.C.); 2Department of Applied Health Science, School of Public Health, Indiana University, Bloomington, IN 47405, USA

**Keywords:** berries, anthocyanins, functional foods, obesity, healthy volunteers, indirect calorimetry, meal tolerance test, insulin sensitivity, glucose tolerance

## Abstract

Obesity in the United States continues to worsen. Anthocyanin-rich fruits and vegetables provide a pragmatic dietary approach to slow its metabolic complications. Given American diet patterns, foods with high anthocyanin content could address dose-response challenges. The study objective was to determine the effect of 100% elderberry juice on measures of indirect calorimetry (IC) and insulin sensitivity/glucose tolerance in a placebo-controlled, randomized, crossover pilot study. Overweight and obese adults were randomized to a 5-week study which included 2 1-week periods of twice-daily elderberry juice (EBJ) or sugar-matched placebo consumption separated by a 3-week washout period. Following each 1-week test period, IC and insulin sensitivity/glucose tolerance was measured with a 3 h meal tolerance test (MTT). Treatment differences were tested with linear mixed modeling. A total of 22 prospective study volunteers (18 F/4 M) attended recruitment meetings, and 9 were analyzed for treatment differences. EBJ was well tolerated and compliance was 99.6%. A total of 6 IC measures (intervals) were created, which coincided with 10–20 min gaseous samplings in-between MTT blood samplings. Average CHO oxidation was significantly higher during the MTT after 1-week EBJ consumption (3.38 vs. 2.88 g per interval, EBJ vs. placebo, *p* = 0.0113). Conversely, average fat oxidation was significantly higher during the MTT after 1-week placebo consumption (1.17 vs. 1.47 g per interval, EBJ vs. placebo, *p* = 0.0189). This was in-line with a significantly lower average respiratory quotient after placebo treatment (0.87 vs. 0.84, EBJ vs. placebo, *p* = 0.0114). Energy expenditure was not different. There was no difference in serum glucose or insulin response between treatments. This pilot study of free-living volunteers describes significant change in IC but not insulin sensitivity with an EBJ intervention. Controlled feeding and increased sample size will help determine the utility of EBJ on these outcomes.

## 1. Introduction

The 2022 estimates from the Centers for Disease Control and Prevention show that the number of states with adult obesity rates at or above 35% has doubled since 2018 [1]. The annual medical costs of obesity in the United States in 2019 was an estimated USD 173 billion [2]. In a press release, the medical authorities associated with the 2022 estimates call for increased access to healthcare, safe places to engage in physical activity, and access to healthy foods if the obesity epidemic is to be slowed or reversed [3]. Within the “healthy foods” category, most Americans fail to eat the recommended servings of fruits and vegetables, many of which contain bioactive food components that may protect against obesity-associated metabolic dysfunction. Within this category are berries. 

Commonly consumed berries are rich in polyphenols, notably flavonoids and anthocyanins [4]. These plant pigments are diverse, and their concentrations vary across berry varieties. Their antioxidant and anti-inflammatory effects are well described [5,6,7]. Moreover, there is a strong body of evidence in preclinical literature that demonstrates a protective effect of berries and their constituent anthocyanins against diet-induced rodent obesity [8,9,10,11,12,13,14,15,16]. Protective effects of anthocyanin consumption are not limited to preclinical data; in prospective cohort studies, anthocyanin consumption is associated with lowered risk of cardiovascular disease and type-2 diabetes, reduced all-cause mortality, and improved weight maintenance [17,18,19,20]. Short-term clinical studies with berry interventions report improvements in glucose tolerance with several different varieties, including blueberries, strawberries, and blackberries [21,22,23,24,25,26,27]. However, we have previously reported mixed results despite a highly controlled experimental design with a 1-week mixed berry intervention [28]. One possible reason for mixed findings could be differences in anthocyanin doses. In an earlier study, improved insulin sensitivity and increased fat oxidation were observed in participants who consumed 360 mg of anthocyanins per day for 1 week from blackberries [27]. However, the follow-up study did not reproduce improvements in insulin sensitivity with a mixed-berry intervention that provided a similar cohort between 109 and 218 mg anthocyanins per day [28]. Despite the promising observations of the former trial, a 360 mg dose of anthocyanins equates to 600 g (4 cups) of blackberries per day, which is similar in anthocyanin density to other common berries. If berry anthocyanins are indeed beneficial in 360+ mg doses, other varieties must be considered to account for practical serving sizes. Here, we employ the same clinical testing as earlier studies, but with an elderberry juice (EBJ) intervention. 

Despite less commercial attention compared to other conventional berry varieties, clinical research with elderberry interventions is warranted due to their high concentration of anthocyanins; because of the 5-fold higher density, EBJ provides 360 mg of anthocyanins with a 177 mL (6-ounce) serving [29]. 

The objective of this pilot study was to test EBJ consumption for effects on substrate oxidation and insulin sensitivity in overweight or obese adults, who are otherwise healthy, utilizing an investigator-initiated, randomized, placebo-controlled, crossover design. The selected testing protocols were utilized in earlier reports that showed significant changes caused by berry interventions within short-term feeding designs. This pilot trial risks insufficient power to detect treatment differences due to a limited sample size, but the utility offered is the determination of potential directionality of treatment differences as a basis for power calculations for follow-up research. Despite this study’s limited power, we report significant effects of elderberry consumption on substrate oxidation. Moreover, as EBJ is an uncommon yet dense source of anthocyanins, an added utility of this pilot study is to determine its consumer acceptability.

## 2. Materials and Methods

### 2.1. Human Participants

Prospective female and male study participants aged 22–75 years old were recruited from the Spokane (WA, USA) metropolitan area from July 2021 to August 2022 using a rolling recruitment and follow-up approach. Study advertisements were circulated through WSU employee listservs and websites, a neighborhood-based social media platform, and a local newspaper outlet. Interested participants were screened for study eligibility with a health history questionnaire and blood chemistries. Inclusion criteria included BMI > 25 kg/m^2^, as this population increases the likelihood of detecting potential differences in indirect calorimetry (IC) and meal tolerance test (MTT) outcome variables with EBJ treatment. Volunteers were excluded from the study if they were pregnant or intended to become pregnant, were lactating, or had given birth in the last year. Additional exclusions included allergy/intolerance to elderberries, a history of bariatric surgery or malabsorption diseases, restrictive, exclusionary, or fad diet patterns, habitual tobacco use in the preceding 6 months, significant (10%) weight loss or gain in the preceding 2 months, cancer in the preceding 3 years, inflammatory bowel disease or other gastrointestinal issues, blood thinning or other medications which may complicate participant safety or interfere with study outcomes, type-2 diabetes requiring management with prescription medication, fasting blood glucose above 125 mg/dL, or active alcoholism. Qualified participants were screened for unusual diet patterns using the ASA24 from the National Cancer Institute (Frederick, MD, USA). Enrolled participants were instructed to follow their habitual diet and to cease from taking any diet supplements beyond a standard multivitamin, calcium, and vitamin D for the entire 5 weeks of the study, and to not donate blood. They were also provided a food exclusion list of high-polyphenol foods including any and all types of berries, red grapes, cherries, plums, red apples, red cabbage, red radishes, red onions, eggplant, and black and red beans. They were instructed to avoid eating any of the listed foods and their products for the entire 5 weeks of the study. The study protocol was approved by the Washington State University (WSU) Institutional Review Board (Pullman, WA, USA) number 18682, and all participants provided written, informed consent. The study protocol was registered on ClinicalTrials.gov (NCT 05723497). The study was conducted at the department of Nutrition and Exercise Physiology (NEP) of WSU’s Elson S. Floyd College of Medicine in Spokane, WA, USA. 

### 2.2. Study Design and Treatment

The study was randomized, placebo-controlled, and crossed over on two treatments. Each 1-week treatment period was separated by a 3-week washout period, as we have previously reported potential carryover effects of berry interventions on serum insulin in studies with shorter (1-week) washout periods [27]. Participants consumed either 177 mL of 100% elderberry juice (River Hills Harvest, Hartsburg, MO, USA) or a flavor and sugar-matched placebo beverage (PL) daily for 1 week. The 177 mL servings were selected to provide 360 mg of cyanidin-3-glucoside equivalents (C3GE) per day, and the C3GE concentration in the elderberry juice was confirmed with the total monomeric anthocyanin spectrophotometric assay [30]. The PL beverage was prepared by North Carolina State University’s Food Innovation Lab (Kannapolis, NC, USA). Participants were blinded to treatment and sequence. Participants were randomized to sequence 1 (elderberry juice then placebo) or sequence 2 (placebo then elderberry juice) using covariate adaptive randomization by the investigator. Daily doses were divided into two 88.5 mL beverage containers and participants were instructed to consume one container in the morning and one in the evening with their meals. Treatments were consumed offsite as part of the participants’ usual meal schedule. Preceding each 7-day treatment period, study staff prepared 14 4-ounce Nalgene bottles with 88.5 mL treatments. Participants were provided with the seven-day supply and were instructed to store their treatment containers at 4 °C until the time of consumption. They were also provided a daily questionnaire to report test beverage consumption, medication use, and any acute illnesses. Compliance was assessed from returned containers and questionnaires collected on the morning of the 8th day of the respective treatment period, which coincided with their testing day.

### 2.3. Participant Testing with Meal and Exercise Challenges

On the morning of the eighth day of each treatment period, participants reported to NEP’s clinical testing laboratory between 7 and 8 a.m. for combination IC and MTT testing. Following a weigh-in, collection of empty beverage containers, completed daily questionnaires, and a general health check-in, an indwelling catheter was placed in the antecubital vein by a study nurse. Following two baseline blood samplings separated by at least 10 min, participants were given 10 min to consume a challenge meal consisting of 80 g toaster waffles, 80 g pancake syrup, and 177 mL EBJ or PL beverage, in accordance with the preceding week’s treatment assignment. The test meal serving sizes were designed to provide at least 80 g of sugar from whole food sources, which accounted for the sugar content of the respective test beverages (Table 1) [31]. Following successful consumption of the challenge meal, participants were made comfortable on a padded examination table in a recumbent position under a TrueOne 2300 metabolic cart canopy system (ParvoMedics, Provo, UT, USA) to measure respiratory gasses. Simultaneous to IC measurement, blood sampling occurred every thirty minutes from the first bite of the challenge meal for three hours for a total of eight blood samplings, including the two baseline samples. Blood was collected into two 8 mL serum tubes, clotted, centrifuged, aliquoted, and stored at −80 °C until analysis. Frozen serum samples were banked until the end of the study’s enrollment phase, at which time, all samples were submitted to a contract lab (Laboratory Corporation of America) for determination of serum insulin and glucose. Ten participants successfully completed both test days, and half the samples were analyzed by the contract lab per day on two consecutive days at the conclusion of the study. 

Earlier reports show change in IC during exercise due to berry anthocyanin treatments [27,32,33,34]; therefore, we incorporated a low-intensity exercise test into our study protocol in addition to the MTT. Following the final blood sampling and discharge of the antecubital catheter and a 15 to 30 min break, participants walked on a treadmill for 30 min at 3 miles-per-hour while IC was assessed using the same metabolic cart system.

### 2.4. Calculations and Statistics

The ParvoOne metabolic cart measures respiratory gasses to calculate grams of carbohydrate (CHO) and fat oxidized, the respiratory quotient (CO_2_ produced/O_2_ consumed), and energy expenditure in kilocalories. CHO and fat oxidation, energy expenditure, and average RQ are calculated by the metabolic cart and reported on a minute-by-minute basis. During the MTT, respiratory measurement was paused at each 30 min blood draw timepoint. Ten minutes of data preceding the first (30 min) postprandial blood draw were analyzed in interval 1, and 20 min of data preceding each subsequent blood draw (intervals 2–6) were analyzed. A total of six time intervals were collected during the three-hour MTT. Respiratory gasses were measured without interruption during the 30 min treadmill walk.

Seven-point serum glucose and insulin response curves were created from the 3 h MTT for both diet periods: blood sampling occurred at −15, 0, 30, 60, 90, 120, 150, and 180 min from the first bite of the challenge meal; and the −15 and 0 min serum glucose and insulin measures were averaged. Incremental area under the concentration curve (iAUC) was calculated for each curve using the central Riemann-sum. Calculations were automated using SAS version 9.4 (SAS institute, Cary, NC, USA) as previously described [35]. Serum glucose concentration is reported as mg glucose per dL serum, and insulin concentration is reported as µIU insulin per mL serum. Serum glucose iAUC is reported as mg·minute per dL, and serum insulin iAUC is reported as µIU·minute per mL.

Linear mixed models were created to test for statistically significant differences between the EBJ and PL beverage treatments using PROC MIXED repeated measures analysis of covariance in SAS version 9.4. Response variables from each treatment period were repeated on volunteer; the best-fitting covariance structure was selected based on information criteria and visualization of residual plots. Normality of residuals was assessed with the Shapiro–Wilk test, and non-normality was addressed with mathematical transformations. The models built to test the outcome variables of IC from the MTT were repeated on treatment and interval and included main effects for treatment, interval, and sequence, an interaction term for treatment*interval, and included volunteer sex, BMI, and age as covariate terms. Treatment*interval and covariates were removed by backward elimination of non-significant terms. The models built to test the outcome variables of IC from the 30 min treadmill walk were similar, except that models did not include interval or treatment*interval terms (the measurements occurred for a continuous 30 min without interruption). The models built to test the glucose and insulin response curves were repeated on treatment and minute, and included main effects for treatment, minute, sequence, covariates, and a treatment*minute interaction term not subject to backward elimination. Similar models were built to test iAUC for both insulin and glucose; this is a single measure per 3 h period, and therefore is not repeated on time. Volunteers were included as a random-side effect term in all models. All models included the Tukey post-hoc HSD correction when reporting group-wise differences. Data are presented as group means ± SEM, and statistical significance is considered when *p* < 0.05. The de-identified raw data is provided as Appendix A: Postprandial MTT energetics; Appendix A: Exercise energetics; and Appendix A: Insulin Glucose Data.

## 3. Results

The CONSORT diagram for this study is reported in Figure 1. Twenty-two prospective volunteers attended an information meeting, and six were lost to follow-up. Of the remaining 16 volunteers who provided informed consent, 4 later declined to participate in the study, and 12 were screened for eligibility. Two volunteers were excluded for not meeting inclusion criteria, and ten participants were randomized to the study protocol. All 10 participants completed the study protocol. One participant was excluded from analysis due to uncontrolled type-2 diabetes, which was indicated by fasting serum glucose >125 mg/dL on both test days. Twice-daily test beverage consumption compliance, assessed by the daily questionnaire and returned containers, was 99.6%. There were no reports of gastrointestinal distress and both treatment beverages were well tolerated. Baseline characteristics for the 9 included participants are reported in Table 2.

IC measures of fat, CHO, EE, and RQ from the MTT are reported in Table 3 and Figure 2. The IC measures from the exercise challenge are reported in Table 4. A number of technical failures during the MTT created an unbalanced dataset; statistical analyses included all partial datasets. There were significant treatment differences for average RQ and corresponding average CHO and fat oxidation across IC time intervals measured during the MTT; none of the models constructed retained a treatment by interval interaction term. Average interval RQ was significantly higher with the EBJ treatment (0.87 vs. 0.84, EBJ vs. PL, respectively, *p* = 0.0114). This corresponds to a significant increase in average CHO oxidation (3.38 vs. 2.88 g per interval, EBJ vs. PL, respectively, *p* = 0.0113) and a significant decrease in average fat oxidation (1.17 vs. 1.47 g per interval, EBJ vs. PL, respectively, *p* = 0.0189) with EBJ treatment compared to PL. There was no treatment effect in average EE (25.1 vs. 25.7 kcal per interval, EBJ vs. PL, respectively, *p* = 0.6918), but there was a significant sex difference (23.8 vs. 31.2 kcal per interval, females vs. males, respectively, *p* = 0.0251). Comparisons in IC measures were similar for the 30 min treadmill walk, but none of the comparisons were statistically significant. There was no effect of treatment sequence in any of the IC outcomes tested during the MTT or exercise challenges.

Serum glucose and insulin MTT-response curves, as well as corresponding incremental area under the concentration curves, are reported in Figure 3 and Table 5, respectively. Linear mixed models were built to test for treatment differences both on individual time points as well as for iAUC for both serum glucose and insulin. There were no significant treatment differences on individual time points or for iAUC for both serum glucose and insulin.

## 4. Discussion

The objective of this study was to test 100% elderberry juice for effects on indirect calorimetry and glucose tolerance/insulin sensitivity. To strengthen the study design, the EBJ was compared to a custom-made sugar- and flavor-matched PL to allow for participant blinding to treatment, and to test EBJ for potential bioactive effects attributable to plant metabolites while accounting for its energy and macronutrient content. Testing was performed in a randomized, crossover design with a three-week washout period to ensure carryover effects were minimized. This project was informed by previous work in which insulin sensitivity and fat oxidation increased in overweight or obese men following a 7 day, 100% controlled, 40% fat diet with either 600 g of blackberries or an energy-matched control food per day [27]. In the current study, EBJ was selected for its unusually high anthocyanin content compared to other commonly consumed berries. A 360 mg dose of anthocyanins provided by 600 g of blackberries is equivalent to the anthocyanin content in 177 mL of EBJ. If threshold anthocyanin dose is a key component to the health-promoting effects of berries, EBJ offers a pragmatic choice for both the consumer and functional food industry. 

There is promising evidence in the rodent literature from several independent research groups that report the effectiveness of anthocyanins from berries against diet-induced obesity and resultant complications with glucose homeostasis (8–16). The proposed mechanisms of action of anthocyanins are broad and include inhibition of intestinal nutrient absorption, modulation of the gut microbiome, pan peroxisome proliferator-activated receptor (PPAR) activation, and modulation of adenosine monophosphate-kinase (AMPK), and peroxisome proliferator-activated receptor gamma coactivator-1 alpha (PGC1-alpha) molecular pathways related to energy balance and thermogenesis [14,36,37,38,39,40,41,42,43,44]. It is difficult to translate the proposed mechanisms to clinical trials, but clinical evidence demonstrates promotion of glucoregulation by berries and anthocyanins [45,46]. This study did not corroborate a benefit on glucoregulation by anthocyanin-rich EBJ. Limitations that could explain a lack of effect include a small sample size and not controlling background diet in the week leading up to the MTT.

Despite no effect of EBJ on serum glucose and insulin following a meal challenge, it caused a significant increase in average CHO oxidation, which was detectable by a metabolic cart measuring respiratory gasses in parallel to the MTT. Interestingly, increased average CHO oxidation was balanced by a significant decrease in average fat oxidation, while EE was unchanged between treatments. Similar directionality was maintained during a 30 min bout of low-intensity exercise, but without reaching significance. The increased CHO oxidation with the EBJ treatment contrasts the findings from our earlier work with blackberries, where a meal challenge with blackberries caused a significant reduction in insulin response and an increase in fat oxidation compared to placebo [27]. A lack of controlled feeding is the major difference between the two study designs that may explain disparate findings. The current findings may suggest increased metabolic flexibility with EBJ when combined with a high-carbohydrate meal. 

The difference in substrate oxidation between treatments without detectable change in serum glucose and insulin response to the MTT are noteworthy, but there is precedent in other interventions. For instance, 1 month of carnitine supplementation increased CHO oxidation and decreased fat oxidation following an MTT in 11 impaired glucose tolerance volunteers, but without change in concurrent plasma glucose, insulin, or free fatty acids (FFA) [47]. Further, a recent study performed comprehensive metabolic testing in twenty healthy young males with a genetic predisposition for type-2 diabetes after acute melatonin treatment, and captured a similar effect but with fat oxidation; during an intravenous glucose tolerance test, fat oxidation increased without change in serum glucose, insulin, or FFA [48]. In the current study, the elevated CHO oxidation may not be sufficient to alter postprandial glucose excursion over the time intervals measured, or the participants’ homeostatic regulation was sufficient to compensate for the increase through endogenous glucose production. Taken together, the increased CHO oxidation with EBJ in the current study may indicate meaningful improvements in CHO homeostasis, but follow-up studies incorporating controlled feeding and adequate statistical power are needed to confirm our observations. 

Anthocyanin bioavailability has often been reported as limited [49], but a (13)-C tracer study demonstrated that the model cyanidin-3-glucoside is more bioavailable than previously thought [50], and this concept has been expanded to anthocyanins more generally [51]. While native dietary anthocyanins are seldom absorbed intact, these reports provide evidence of both absorption of conjugated metabolites in the small intestine and microbial metabolites in the colon. These metabolites are reportedly detectable in serum samples after 30 min [50], and this rapid absorption of anthocyanin metabolites is one provisional reason why effects on CHO oxidation were detected in this study. It should be noted that the anthocyanin profiles between elderberry juice, blackberries, and other commonly consumed berries are heterogeneous [28,52,53]. These differences potentially impact the bioactivity of the respective berry. Future research measuring physiological responses similar to this study between different berries is warranted to further elucidate differences in bioactivity.

Major strengths of this study include a robust crossover design, which was informed by earlier work, to incorporate an adequately spaced washout period and work with a food sciences research institute to prepare a flavor and sugar-matched placebo beverage. The 3-week washout period was informed by earlier work that suggested carryover effects on insulin sensitivity were possible with a shorter 1-week washout period; none of our linear models detected a significant sequence effect in this project. The decision to work closely with a food science institute to prepare a placebo was to account for participant bias and potential placebo effects by blinding treatment, which is a major challenge of nutritional interventions. Major weaknesses of this pilot study include a small sample size, and a lack of controlled feeding prior to test days. The limited sample size was driven primarily by the challenges surrounding the COVID-19 pandemic, and because this was a pilot study with limited resources, controlled feeding was not possible. 

To our knowledge, this is the first study to perform meal tolerance and calorimetry testing of high-anthocyanin 100% elderberry juice. This unique and nascent commodity [54] is a rich source of anthocyanins, with a 5-fold higher content compared to common berry sources. Interestingly, elderberries have suffered bad press since a 1980s-era CDC case report alluded to potential acute toxicity, probably due to cyanogenic glycosides present in the plant’s stems and seeds [55]. The safety profile of EBJ has been described recently [56]. This study corroborates the safety of commercially available EBJ for human consumption, as no adverse effects were reported after 10 participants consumed 6-ounces of juice per day for 1-week or following a 6-ounce serving in a single sitting. Elderberry products are widely marketed, but, as of 2010, annual sales only constituted USD 6.8 million in the United States [57].

## 5. Conclusions

In conclusion, 7-day feeding of 100% elderberry juice increases postprandial CHO oxidation following a high-sugar MTT but does not affect serum glucose or insulin in a small sample of overweight or obese free-living participants. Elderberry juice is well tolerated, and follow-up work with more robust designs, controlling background diet with adequate statistical power, and testing similar glucoregulatory and calorimetry measures are warranted to confirm these potentially beneficial outcomes. 

## Figures and Tables

**Figure 1 nutrients-15-02072-f001:**
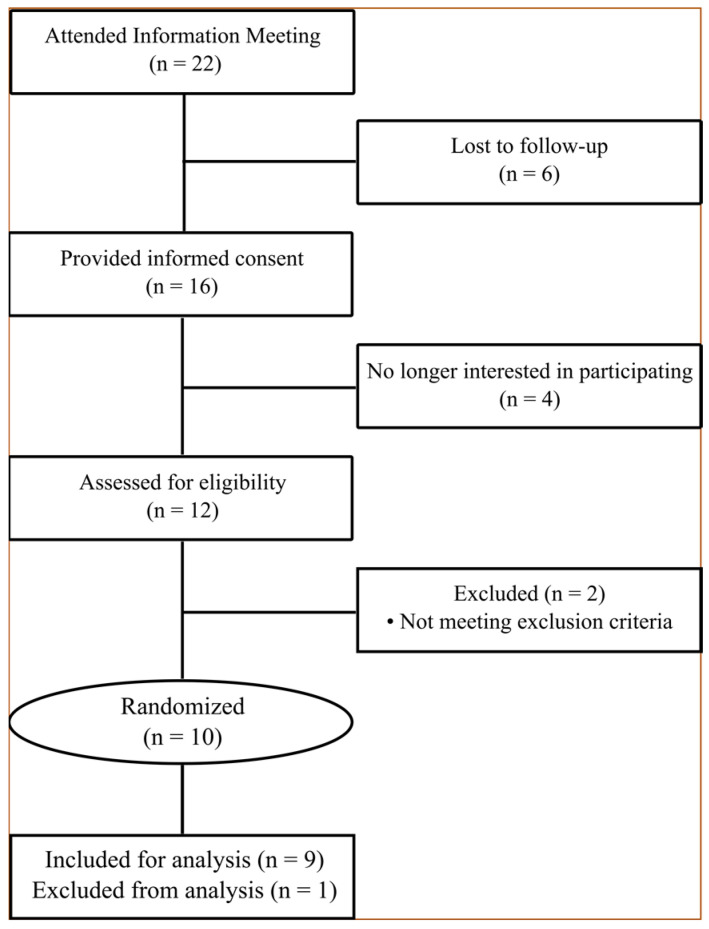
CONSORT (Consolidated Standards of Reporting Trials) diagram. One participant was excluded from analyses due to uncontrolled type-2 diabetes.

**Figure 2 nutrients-15-02072-f002:**
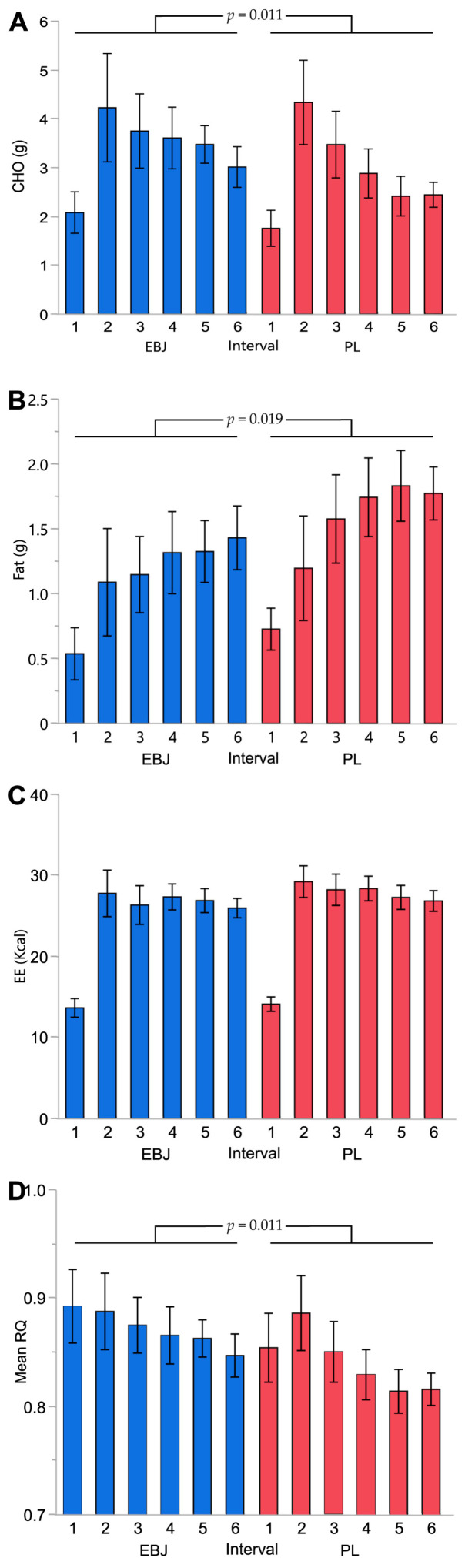
MTT Energetics. Postprandial substrate oxidation, energy expenditure, and respiratory quotient for six intervals following a meal challenge. (**A**) EBJ increased mean CHO oxidation, (**B**) reduced mean fat oxidation, (**C**) did not change EE, and (**D**) elevated mean RQ compared to PL. Intervals 1–6 represent minutes 20–29, 40–59, 70–89, 100–119, 140–159 following a morning meal, respectively. Error bars represent SEM. EBJ intervals 1 and 2 *n* = 6, and intervals 3–6 *n* = 8. PL intervals 1–8 *n* = 7. MTT, meal tolerance test; CHO, carbohydrate oxidation (g); Fat, fat oxidation (g); EE, energy expenditure (Kcal); RQ, respiratory quotient.

**Figure 3 nutrients-15-02072-f003:**
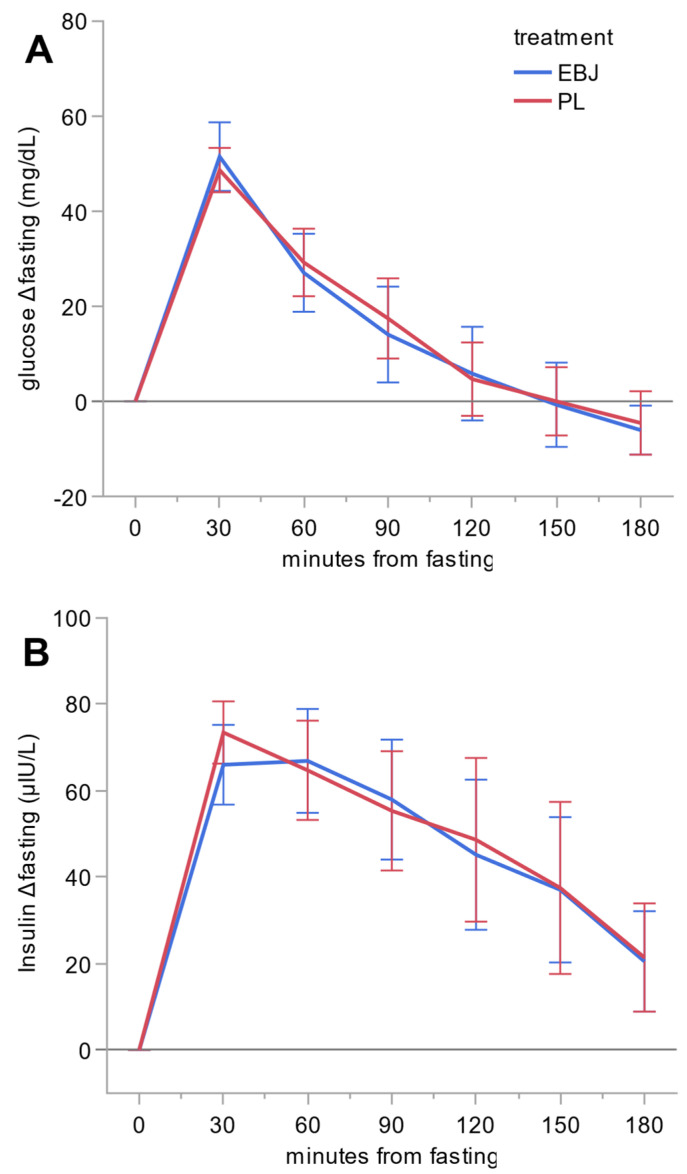
Participants’ serum glucose (**A**) and insulin (**B**) during responses to a meal challenge. No differences in time by treatment or total iAUC were found for either glucose or insulin. *n* = 9.

**Table 1 nutrients-15-02072-t001:** Macronutrient and energy content of the meal-based tolerance test.

Item	Quantity (g)	Protein (g)	CHO (g)	Sugar (g)	Fat (g)	Energy (kcals)
Waffle	80	4.6	34.3	4.6	5.7	205.7
Syrup	80	-	72	58.7	-	293.3
Test beverage	177	-	19.5	19.5	-	78
Total	337	4.6	125.8	82.7	5.7	577

CHO, carbohydrate.

**Table 2 nutrients-15-02072-t002:** Baseline participant characteristics.

Characteristic	Mean ± SEM
*n* (sex)	7 (f) 2 (m)
Age	55.3 ± 3.0
Weight (kg)	83.7 ± 4.7
BMI (kg/m^2^)	29.8 ± 1.4
Waist Circumference (cm)	99.2 ± 3.1
Total cholesterol (mg/dL)	199.2 ± 7.4
LDL (mg/dL)	113.1 ± 5.4
HDL (mg/dL)	66.4 ± 5.9
TG (mg/dL)	110.9 ± 12.3
Systolic Blood Pressure (mm Hg)	120.1 ± 3.1
Diastolic Blood Pressure (mm Hg)	77.7 ± 2.6
Glucose (mg/dL)	93.1 ± 2.7

Total cholesterol, LDL, HDL, TG, and glucose tests are measured in 12-h fasted serum samples.

**Table 3 nutrients-15-02072-t003:** Postprandial substrate oxidation, EE, and mean RQ for overweight and obese adults following a meal challenge with EBJ or PL.

	EBJ	PL	
	Mean ± SEM	Mean ± SEM	*p*
CHO (g)	3.37 ± 0.26	2.88 ± 0.24	0.011
Fat (g)	1.17 ± 0.12	1.47 ± 0.13	0.019
EE (Kcal)	25.1 ± 0.99	25.7 ± 1.01	0.692
Mean RQ	0.869 ± 0.010	0.841 ± 0.011	0.011

EBJ, elderberry juice; PL, placebo; CHO, carbohydrate oxidation; Fat, fat oxidation; EE, energy expenditure; RQ, respiratory quotient. EBJ *n* = 8 (two partial); PL *n* = 7. Four participants had complete datasets, where all six intervals were successfully measured from both test days. Four participants had partial datasets, where all six intervals were successfully measured from at least one of the two test days. Finally, 1 participant had a near-complete dataset, where 1 test day had all 6 intervals measured, and the other had intervals 3–6 of the 6 intervals measured. Comprehensive information on missing data is provided in the Appendix A.

**Table 4 nutrients-15-02072-t004:** Substrate oxidation, EE, and RQ for participants who walked on a treadmill for 30 min at 3 mph after consuming EBJ or PL for one week.

	EBJ	PL	
	Mean ± SEM	Mean ± SEM	*p*
CHO (g)	21.3 ± 2.54	18.1 ± 2.33	0.107
Fat (g)	7.60 ± 1.10	8.38 ± 1.19	0.333
EE (Kcal)	160.2 ± 11.1	154.2 ± 10.5	0.238
RQ	0.864 ± 0.014	0.847 ± 0.015	0.239

EBJ, elderberry juice; PL, placebo; CHO, carbohydrate oxidation; Fat, fat oxidation; EE, energy expenditure; RQ, respiratory quotient. *n* = 9.

**Table 5 nutrients-15-02072-t005:** The 180 min serum glucose and insulin iAUC in overweight and obese adults following a meal challenge.

	EBJ	PL	*p*
Glucose iAUC (mg·minute per dL)	3233 ± 566	3972 ± 1043	0.317
Insulin iAUC (µIU·minute per mL)	8504 ± 2073	8714 ± 2042	0.559

iAUC, incremental area under the curve; EBJ, elderberry juice; PL, placebo. *n* = 9.

## Data Availability

Deidentified data described in the manuscript was uploaded to Nutrients upon article submission and is freely available to the public without restriction.

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
