# Peer review of "One-Week Elderberry Juice Treatment Increases Carbohydrate Oxidation after a Meal Tolerance Test and Is Well Tolerated in Adults: A Randomized Controlled Pilot Study"

_nutrients, 2023, doi:10.3390/nu15092072_

Round 1

Reviewer 1 Report

The study carried out is very interesting, mainly because of the method you have used. I would recommend to repeat the study with a statistically significant sample in order to reduce the errors. 

Reviewer 2 Report

English revision should be performed.

Mayor comments:

My main concern is how do you control the rest of the diet? Do you advice participant to limit the intake of other anthocyanins-rich foods? or do you gave them a menu?

- Inclusion criteria include malabsorption diseases, please specify if you exclude autoinmune diseases or other diseases and the total amount of selected participants excluded for each reason.

- This beverage only contain anthocyanidins? Do you know the total polyphenol amount in the beverage? other flavonoids?

- In the methods section 2.4. Please include respiratory gases measurements and glucose/insulin curves in measurements and create a specific section for statistical analysis.

- Figure 1 should include how many participants were assessed to participate and the reason why they were excluded.

- In table 2 more measurements should be included if posible, such as waist circumference, smoking status, physical activity, supplementation use, history of familiar CVD, medication use (mainly CVD related drugs), etc.

- Discussion: data about the intake of this product in general population should be included... Is this juice "traditional food"?

Minnor comments:

- In the introduction section you define berries in line 54-55. Please move it to the first "berries" in the text.

- Line 86 you mentioned "Spokan metropolitan area", please add country.

- In table 1 you mentioned "Test beverage". Please specify if this is the juice.

Reviewer 3 Report

This study looks at the effects of consuming elderberry juice on indirect calorimetry and postprandial insulin and glucose.

However, the paper is not well written and there are issues that need addressing:

-          “Inclusion criteria included BMI > 25 kg/m2 as this population increases the likelihood of detecting potential differences in indirect calorimetry (IC) and meal tolerance test (MTT) outcome variables with EBJ treatment.” It is unclear what the purpose of the study is. Is it to see if elderberry juice can protect from weight gain, promote weight loss or protect from obesity related metabolic disease? Or are the authors just using obese patients as a model to test IC and IS?

-          Another issue that the authors have rightfully indicated is sample size and missing data. It is understandable that due to unforeseen circumstances challenges have occurred in data collection. However, more detail on what was analysed and the implications of the missing data, need to be addressed: It is indicated from line 272-273 that not all the datapoints/participants have been analysed for all the markers measured. The implications of missing data and imbalances needs to be addressed first by clearly stating what datapoints and participants were analysed and for which treatment and why are their missing data (this could be presented in a table under the consort diagram). Then the authors need to address how this affected the power of the study and whether the imbalance between groups has influenced the results. For example, if participants with missing data were included in the IC analysis, how does this compare if the participants with missing data have been excluded from analysis? Also, if no data from the insulin and glucose were missing, could this explain why authors have found increased CHO oxidation but no difference in postprandial glucose?

There seems to be an assumption from the authors that anthocyanins are the main players in the effects seen in their previous blackberry study. However, blackberries have other important phenolic compounds and the unique compound(s) and their combinations, found in blackberries might be a more plausible reason for why there was no effect on insulin in the current study and the previous mixed berry study.

Another implication for discussion is that this study was done as part of the participant’s normal diet, whereas previous studies used controlled diets. Authors should comment on how likely such an intervention would benefit people in a real-world scenario. That is, is there any point of consuming berries or berry juice in the way recommended by the authors, if it will have no effect when consumed as part of normal diet?

 More specific comments:

-          Line 20-21: Only need to mention the number of participants analysed.

-          Line 36, what is CDC?

-          Line 55 onwards. Need better explanation and detail of the previous studies.

-          Line 72-73. Not clear why these markers were chosen in relation to obesity.

-          Line 86. What is rolling recruitment and follow up basis?

-          Line 123: which covariates were monitored for this randomization? Was this due to the small number of male participants?

-          Line 154-155: Why were only half the samples analysed per day and which ones?

-          Need a comparison between EBJ and blackberry flavonoids (the compounds identified in the authors’ previous blackberry study ).

-          Table 2: are these fasting levels? If so, need to mention.

-          Line 283-284. Not appropriate statement as authors only had 2 male participants enrolled in the study.

-          Table 3. Is this raw data or adjusted against baseline?

-          Line 317. There are other important bioactives in elderberry not just anthocyanins.

-          Need to discuss how much of the bioactives in whole elderberry will be affected when consumed in juice form. Is juice the same as whole berries?

-          Line 325-326 need to provide references and which anthocyanins are higher in elderberry compared to blackberries? How do they compare in cyanidin-3-O-β-glucoside?

-          Line 352-353. Not clear, needs to be re-written.

-          Line 393. What is “it”?

Round 2

Reviewer 2 Report

Thank you for considering my comments and excuse me for the English editing comment. 

Author Response

Thank you for helping us to improve the quality of this manuscript.